**Data Availability Statement:** Hong Kong data cannot be shared publicly because of ethics

# Main and interacting effects of physical activity and sedentary time on older adults' BMI: The moderating roles of socio-demographic and environmental attributes

**Delfien Van Dyck**[1,2]*, **Anthony Barnett**[3], **Jelle Van Cauwenberg**[2,4], **Casper J. P. Zhang**[5], **Cindy H. P. Sit**[6], **Ester Cerin**[3,5,7]

1 Faculty of Medicine and Health Sciences, Department of Movement and Sports Science, Ghent University, Ghent, Belgium, 2 Research Foundation Flanders, Brussels, Belgium, 3 Mary MacKillop Institute for Health Research, Australian Catholic University, Melbourne, Australia, 4 Faculty of Medicine and Health Sciences, Department of Public Health and Primary Care, Ghent University, Ghent, Belgium, 5 School of Public Health, The University of Hong Kong, Hong Kong, China, 6 Faculty of Education, Department of Sports Science and Physical Education, The Chinese University of Hong Kong, Hong Kong, China, 7 Baker IDI Heart and Diabetes Institute, Melbourne, Australia

* delfien.vandyck@ugent.be

## Abstract

### Objectives

Our first aim was to examine the main and interacting effects of accelerometer-based sedentary time (ST) and moderate-to-vigorous physical activity (MVPA) with BMI and the likelihood of being overweight/obese in Hong Kong and Ghent (Belgium) older adults. Second, we examined whether these main associations and interactions between MVPA and ST were moderated by socio-demographics (gender, education) and objective neighbourhood attributes supposed to be associated with healthy food intake (food outlet density, neighbourhood-level SES). Finally, we determined whether the associations and interactions were generalisable across study sites.

### Methods

Data from the ALECS (Hong Kong) and BEPAS Seniors studies (Ghent), two comparable observational studies, were used. Older adults (n = 829, 65+) provided self-reported socio-demographic information and objective MVPA and ST data using Actigraph accelerometers. Annual household income data and GIS software were used to assess neighbourhood-level SES and food outlet density. Generalised additive mixed models were conducted in R.

### Results

ST was linearly and positively related to both weight outcomes in the overall sample, while MVPA was not. The overall-sample analyses including the two-way interaction between MVPA and ST showed no interactions between these behaviours on weight outcomes. Three site-specific findings were identified, showing distinct associations in Hong Kong

restrictions, as the participants consented to provide their data to the project investigators only. Data are available from the University of Hong Kong (contact via Ester Cerin; ecerin@hku.hk) for researchers who meet the criteria for access to confidential data upon approval of the Human Research Ethics Committee of the University of Hong Kong (hrec_data@hku.hk). Belgian data are publicly available. All relevant Belgian data are within the manuscript and its Supporting Information files.

**Funding:** The Hong Kong study received a General Research Fund grant from the University Grant Committee (Hong Kong) (HKU 741511H). EC is supported by an Australian Research Council Future Fellowship (FT3 #140100085). DVD and JVC are supported by Research Foundation Flanders (FWO). The funding bodies had no input in study design and the collection, analysis and interpretation of data and the writing of the article and the decision to submit it for publication. All authors are independent from the funding bodies.

**Competing interests:** The authors have declared that no competing interests exist.

compared to Ghent. Study site moderated the interaction between ST and MVPA on both weight outcomes, the interaction between education and ST on both weight outcomes and the interaction between the number of food outlets and ST on being overweight/obese.

## Conclusions

The country-specific effects confirm the cultural dependency and complexity of the associations between PA, ST and weight outcomes. Future longitudinal international studies including older adults from multiple regions assessing PA, ST, weight outcomes and dietary intake should be encouraged. Such studies are needed to refine the recommendations regarding ST and PA in older adults in light of preventing overweight and obesity.

## Introduction

In the last decades, the prevalence of overweight and obesity has increased dramatically in developed and most developing countries worldwide [1,2]. Also, evidence is clear that the global prevalence of overweight and obesity is alarmingly high in older adults [3,4]. This 'obesity pandemic' has severe consequences for the physical and mental health of older adults, including increased rates of chronic diseases such as type 2 diabetes and cardiovascular diseases, decreased cognitive functioning, decreased quality of life, higher rates of depression and increased health care costs [5–7]. Consequently, it is of utmost importance to identify the factors associated with overweight and obesity in older adults in order to prioritize prevention strategies.

Older adults are the population subgroup engaging in the highest levels of sedentary time (ST) [8] and the lowest levels of PA [9]. A large body of literature shows that achieving recommended levels of moderate-to-vigorous PA (MVPA) is negatively associated with markers of overweight and obesity (e.g. BMI, waist circumference) in older adults [10,11]. Furthermore, growing evidence documents a positive association between sedentary behaviours (mainly TV viewing) and the odds of overweight and obesity [12–14]. So, based on this evidence, it seems that interventions targeting PA and ST in older adults are key to prevent overweight, obesity and their related health consequences. Nonetheless, the currently available studies have some shortcomings and specific research questions remain to be answered.

First, most previous studies only included self-reported measures of ST, often limited to TV viewing [15]. It is known individuals tend to underreport socially undesirable behaviours [16] and more specifically it has been shown that older adults under-report their levels of sedentary behaviour [17]. Furthermore, one needs to be cautious when using TV viewing as a proxy for total ST, as the level of agreement between both seems to be limited [18]. Consequently, the use of objective measures of ST is strongly encouraged to examine the associations with health outcomes like overweight and obesity.

Second, although there is ample evidence for direct associations of PA and ST with weight outcomes, evidence regarding the interaction effects of both behaviours on weight-related outcomes is not straightforward. To our knowledge, very few studies have examined such interactions in older adults [15], and studies in adults show mixed results. On the one hand, some studies have found that high levels of PA may ameliorate the negative effects of ST on obesity [19,20] or that the association between ST and weight outcomes becomes non-significant once PA levels are included in the statistical models [21,22]. On the other hand, some studies found clear associations of ST with weight outcomes like body fat mass, irrespective of

the levels of PA [23]. A more profound understanding of these interactions in older adults is needed to determine the focus of future behaviour change interventions. For example, reducing ST may be a more realistic long-term health behaviour goal for older adults facing physical limitations than increasing MVPA. If ST appears to be a determinant of overweight and obesity even at low levels of PA, targeting ST in interventions in older adults might be a promising strategy.

Third, little is known regarding the role of potential moderators in the associations of PA and ST with weight outcomes. Next to other factors (e.g. differences in measurement methods), the presence of moderators might explain some of the previous inconsistent interactions identified between PA and ST with regard to weight outcomes. Only some evidence regarding the moderating effect of gender is available from previous studies. For example, Suminski and colleagues [19] studied the moderating role of gender in the associations of PA and ST with percentage body fat, and found ST was positively associated with percentage body fat only in women who did not meet the PA guidelines. No such associations were identified in men who were insufficiently active. Similarly, in an international study including adults from 10 geographically and culturally diverse countries, ST was curvilinearly associated with BMI in women from the USA sites included in the study, but not in men or in adults from the other 9 countries [24]. Such differences in associations between men and women might be explained by the complex nature of ST and PA and the differences in prevalence of these behaviours according to gender [25].

Next to gender, a more thorough understanding is needed on the moderating effects of other socio-demographic variables (e.g. age, socio-economic status) that might impact the associations between PA, ST and weight outcomes. To our knowledge, the moderating role of these factors has not previously been examined in older adults. In addition, following the socio-ecological framework of health behaviours [26] that emphasizes the interplay between individual-level variables, physical environmental factors and health behaviours, physical environmental factors might also be interesting to examine as potential moderators of the association of PA and ST with overweight and obesity. More specifically, because BMI is largely dependent on energy intake [27], physical environmental moderators that are associated with food intake may attenuate or strengthen the relation of PA and ST with weight outcomes and should be studied. A review study of Rahmanian et al [28] concluded that, although most associations between the built environment and food intake were inconsistent because of methodological challenges, access to healthy food options in the neighbourhood (e.g. food stores, local farmer market) was associated with a higher consumption of fruit and vegetables. Furthermore, living in a low-income neighbourhood was associated with lower access to healthy food options, probably resulting in a less healthy food intake [29].

Considering the shortcomings of the current literature, we used data of two comparable cross-sectional studies conducted in older adults in Hong Kong and Ghent (Belgium). Including data from these geographically and culturally diverse study sites is a particular strength as it allows testing for the generalisability of findings across countries. The first aim of this study was to examine the main and interacting effects of accelerometer-assessed ST and MVPA with BMI and the likelihood of being overweight/obese in older adults. The second study aim was to examine whether these direct associations and interactions between MVPA and ST were moderated by socio-demographic characteristics (gender, education) and objective neighbourhood attributes that are supposed to be associated with healthy food intake (food outlet density, neighbourhood-level SES). Finally, we determined whether the associations and interactions studied above were generalisable across study sites (Hong Kong vs Ghent).

## Materials and methods

### Procedures and participants

Data from two observational cross-sectional studies with comparable protocols and measures were used in this paper: the Active Lifestyle and the Environment in Chinese Seniors (ALECS; first wave) [30] and Belgian Environmental Physical Activity Study in Seniors (BEPAS Seniors) [31]. Hong Kong data collection took place from October 2012 to December 2014 (for the data relevant to this study), Ghent data collection from October 2010 to September 2012. The ALECS project was approved by the ethics committee of the Department of Health (Hong Kong Special Administrative Region, HKSAR) and the University of Hong Kong Research Ethics Committee for Non-Clinical Faculties (EA270211). The BEPAS Seniors study protocol was approved by the Ethics Committee of the Ghent University Hospital (B670201423000).

Details regarding neighbourhood selection and participant recruitment of ALECS and BEPAS Seniors have been described elsewhere [30,31]. In both Hong Kong and Ghent, participants were equally recruited across four types of neighbourhoods, based on neighbourhood-level socioeconomic status (SES; high versus low) and objective neighbourhood-level walkability (Geographic Information System (GIS)-based index of dwelling density, intersection density and land use mix; high versus low). Ghent participants were selected from 20 neighbourhoods, each consisting of 1 to 5 adjacent statistical sectors, the smallest administrative units for which socio-demographic information is available [31]. Hong Kong participants were selected from 124 Tertiary Planning Units, the smallest administrative units with publicly-available census-level data [30].

In Ghent, 1750 community-dwelling older adults (≥65 years) were randomly selected from the statistical sectors and received an information letter through mail with a follow-up home visit approximately one week after receiving the letter. Inclusion criteria were being able to understand and speak Dutch, living independently and able to walk 200 meters without severe physical restrictions. In total 1260 potential participants were found at home at the follow-up home visit. Of those, 125 were not eligible to participate and 508 participated in the study. In total, 427 participants provided all data necessary for the current analyses. In Hong Kong, residential addresses or other contact data could not be directly obtained because of the Personal Data (Privacy) Ordinance [32]. Consequently, participants (≥65 years) were recruited in person in Elderly Health Centres of the Health Department (two thirds) and from elderly community centres close to the selected neighbourhoods (one third). Inclusion criteria were being able to understand and speak Cantonese, living in one of the selected neighbourhoods for at least six months, being able to walk unassisted for at least 10 meters and being cognitively intact (Mini-Mental State Examination Score >22). Of the 1602 potential participants initially contacted, 322 were ineligible and 909 participated in the study. Approximately 45% of the participants were randomly selected to wear an accelerometer to objectively assess their PA and ST. Of these 416 participants, 402 provided valid accelerometer data and could be included in the current analyses. So, in total 829 older adults (402 Hong Kong and 427 Ghent) were included in the analyses. All participants in Ghent and Hong Kong provided written informed consent, and a trained interviewer collected data in their homes and health/community centres, respectively.

### Measures

**Dependent variable: BMI.**    BMI ($kg/m^2$) was calculated using objectively-assessed weight and height. In Ghent, a SECA 214 stadiometer (0.1 cm) and a SECA 813 Robusta weight scale (0.1 kg) were used. In Hong Kong, the height and weight of each participant were measured

using a portable stadiometer Invicta model 2007246 (0.1 cm) and Tanita BC-541 scales (0.1 kg). Before these measurements, participants were asked to remove their shoes and only wear light clothing. BMI was treated as both a continuous and categorical variable. Due to the relatively small number of sample participants in the obese and underweight categories, BMI values were categorised as normal/underweight or overweight/obese. For Ghent, BMI categorisation was $\leq 24.9$ kg/m$^2$ for normal/underweight and $\geq 25$ kg/m$^2$ for overweight/obese). For Hong Kong, the categorisation was $\leq 22.9$ kg/m$^2$ and $\geq 23$ kg/m$^2$ respectively [33].

**Independent variables: Accelerometer-assessed MVPA and ST.**   In both study sites, MVPA and ST were objectively assessed using Actigraph accelerometers (Fort Walton Beach, FL, USA; GT3X in Hong Kong, GT3X and GT3X+ in Ghent). Accelerometers are valid and reliable tools to assess PA and ST in older adults [34,35]. Participants were asked to wear the accelerometer above the right hip bone for seven consecutive days during waking hours, and to remove it only for water-based activities and to sleep. Data were collected using 60-second epochs, as in previous studies in older adults [36,37]. Only data capturing the vertical axis were included for analysis in the present study. A valid day was defined as a day with at least 10 hours of recorded activity, and periods with $\geq 90$ minutes of consecutive zeros were categorized as non-wear time [38]. Data were included in the analyses when a participant had at least five valid days, including at least one weekend day. Activity categories were defined with the cut points of Barnett and colleagues [39] and of Aguilar-Farias and colleagues [40] both developed specifically for older adults and using the Low Frequency Extension filter, as in the current study. Values <25 counts/min were categorised as ST [40] and values $\geq$1013 counts/min were categorised as MVPA [39].

**Covariates and moderators.**   Age, gender, education (primary school versus higher), marital status (married or cohabiting versus widowed or other), neighbourhood-level SES (high versus low) and study site (Hong Kong or Ghent) were included as covariates in the analyses of the main effects of ST and MVPA on BMI. All these variables, except for marital status were also considered as moderators of ST and MVPA associations with BMI. The socio-demographic information (age, gender, education and marital status) was assessed with an interview-administered questionnaire in both study sites [30,31]. Neighbourhood-level SES was assessed using census-based median annual household income data.

Objectively-assessed food outlet density was included as a moderator in the regression models. Objective food outlet density was defined as the number of food outlets/km$^2$, assessed in a 1-km street network buffer around each participant's residential address using GIS. Because increases in dwelling density lead to increases in food outlet density and dwelling density is potentially related to BMI [41], objective dwelling density was included as an additional covariate (potential confounder) in the models with food outlet density as a moderator. Objective dwelling density was defined as the number of dwellings/km$^2$, assessed in a 1-km street network buffer using GIS.

## Data analysis

Descriptive statistics of sample characteristics (participant demographics, objective neighbourhood attributes, activity levels, and BMI) were computed. We also tested the significance ($p < 0.05$) of the differences in these characteristics between the two samples using, as appropriate, generalised linear models and multinomial logistic regression with robust standard errors accounting for clustering at the administrative unit level.

To account for clustering at the administrative unit level and the possibility of curvilinear associations, generalised additive mixed models (GAMMs) [44] were used to estimate confounder-adjusted associations of ST and MVPA with BMI and the interaction effects of ST

and MVPA on BMI [24]. We also examined whether these associations and interactions were moderated by gender, education, neighbourhood-level SES, food outlet density and study site by including 2- and 3-way interaction terms in the GAMMs, as appropriate. Separate sets of GAMMS for continuous BMI and BMI categorised as normal/underweight or overweight/ obese were estimated. Gaussian variance and identity link functions were used to model associations of ST and MVPA with continuous BMI data, whereas binomial variance and logit link functions were used to model associations of ST and MVPA with BMI as a categorical variable [42]. Statistically significant (p<0.05) 3-way interactions were further investigated by first probing 2-way interactions (e.g., ST by food outlet density interaction) at meaningful values of the third interaction-term variable (e.g., Ghent and Hong Kong for study site). Then, if a 2-way interaction term indicated at least a weak moderating effect (p<0.10) [43] (e.g., ST by food outlet density interaction effect in Ghent), the 2-way interaction was probed by estimating the associations of the activity variable (e.g., ST) with BMI in the specific subgroup of participants (e.g., those living in Ghent) at meaningful values of the moderator (e.g., ST associations with BMI at mean-1SD, mean and mean+1SD values of food outlet density in participants from Ghent). Curvilinearity of associations was tested by comparing the Akaike Information Criterion (AIC) values of GAMMs with linear regression terms for ST and MVPA versus GAMMs with ST and MVPA modelled using thin plate smooth terms [44]. Evidence of curvilinearity was deemed sufficient if the model with smooth terms yielded a > 5-unit smaller AIC than the model with linear effects [45]. Analyses were conducted in R version 3.6.1 (https://www.R-project.org) using the packages 'mgcv [46], 'gmodels' [47], and multcomp [48].

## Results

### Participant characteristics

Participant characteristics are presented in Table 1. Mean age was 75 years and slightly higher in the Hong Kong sample ($p$ = 0.025). The total sample (n = 829) consisted of 61% female participants, 69% and 54% in Hong Kong and Ghent, respectively ($p$<0.001). The sample from Ghent had a higher level of education than the Hong Kong sample ($p$<0.001), with 75% versus 47% having an educational level higher than primary school. Having a car in the household was much more prevalent in Ghent (81%) than in Hong Kong (29%) ($p$<0.001). The mean ST per day was 414 minutes, with the Ghent cohort accumulating an average of 16 more minutes per day than the Hong Kong cohort ($p$ = 0.795). For the total cohort, the average MVPA was 56 minutes per day, with Hong Kong participants engaging in approximately 26 minutes more per day than their Ghent counterparts ($p$<0.001). Mean BMI was 23 kg/m$^2$ and, on average, ~ 1kg/m$^2$ lower in Ghent ($p$<0.001). Because of the different cut-off values for categorising overweight/obesity in Asian and European older adults [33], the distribution of BMI categories was very different across sites with 57% of those in Hong Kong being overweight/obese as compared to 24% in Ghent ($p$<0.001).

### Associations of ST, MVPA and co-variates with BMI and being overweight/ obese (Table 2)

While the associations of MVPA with BMI and being overweight/obese were not significant, those of ST were positive, linear and strong. On average, compared to males, females had a significantly lower BMI and were less likely to be overweight/obese. High neighbourhood SES was associated with a lower BMI. Study site was a significant correlate of BMI and being overweight/obese, with participants from Ghent having lower BMI and being less likely to be classified as overweight/obese than the participants from Hong Kong.

**Table 1. Sample characteristics.**

|  | ALECS & BEPAS (n = 829) | ALECS (n = 402) | BEPAS (n = 427) | p-value (difference between samples) |
|---|---|---|---|---|
| **Participant demographics** |  |  |  |  |
| Age (years; mean ± SD) (n = 828) | 74.83 ± 6.18 | 75.55 ± 6.15 | 74.16 ± 6.14 | 0.025 |
| Gender (% female) (n = 828) | 61.35 | 68.91 | 54.10 | <0.001 |
| Education (% above primary school) (n = 829) | 61.04 | 46.77 | 74.47 | <0.001 |
| Marital status (% married or cohabiting) (n = 828) | 64.66 | 62.94 | 66.28 | 0.284 |
| Car in the household (% yes) (n = 827) | 55.49 | 28.86 | 80.56 | <0.001 |
| Neighbourhood SES (low/high, % low) (n = 828) | 51.15 | 50.75 | 51.52 | 0.951 |
| **Objective neighbourhood attributes (mean ± SD, n = 829) (GIS)** |  |  |  |  |
| Dwelling density (1 km buffer, households/km$^2$) | 21,545.64 ± 21,210.69 | 36,316.04 ± 21,489.54 | 7,640.018 ± 6,328.31 | <0.001 |
| Food outlet density (1 km buffer, outlets/km$^2$) | 39.73 ± 52.22 | 26.41 ± 15.17 | 52.27 ± 68.98 | 0.103 |
| **Measures of activity (n = 829)** |  |  |  |  |
| Sedentary time (average minutes/day) | 414.13 ± 98.01 | 404.94 ± 98.11 | 422.78 ± 97.25 | 0.795 |
| MVPA (average minutes/day) | 55.78 ± 37.50 | 69.00 ± 36.31 | 43.33 ± 34.24 | <0.001 |
| **Measures of weight status (n = 820)** |  |  |  |  |
| BMI (kg/m$^2$) | 22.96 ± 3.58 | 23.49 ± 3.34 | 22.46 ± 3.73 | <0.001 |
| BMI categorised (%) |  |  |  | <0.001 |
| Normal or underweight | 59.88 | 43.28 | 75.84 |  |
| Overweight | 30.85 | 41.79 | 20.33 |  |
| Obese | 9.27 | 14.93 | 3.83 |  |

SES = socioeconomic status; SD = standard deviation; GIS = Geographic Information Systems; MVPA = moderate-to-vigorous physical activity; BMI = body mass index; BMI categories in Ghent: normal/underweight < 25 kg/m$^2$, overweight 25—≤ 29.9 kg/m$^2$, obese ≥ 30 kg/m$^2$; BMI categories in Hong Kong: normal/underweight < 23 kg/m$^2$, overweight 23—≤ 26.9 kg/m$^2$, obese ≥ 27 kg/m$^2$.

## Moderating effects (2- and 3-way interactions) on associations of ST and MVPA with BMI and being overweight/obese (Table 3)

Only a few significant moderators of ST or MVPA associations with BMI or being overweight/obese were found. Study site moderated the interaction between ST and MVPA on both BMI and being overweight/obese. Specifically, a significant ST by MVPA interaction on BMI was found in Hong Kong only. In Hong Kong, the positive association between ST and BMI was

**Table 2. Associations of ST, MVPA and co-variates with BMI and being overweight/obese.**

|  | BMI (kg.m$^{-2}$) | | | Being overweight/obese | | |
|---|---|---|---|---|---|---|
|  | **b** | **95% CIs** | **p-value** | **OR** | **95% CIs** | **p-value** |
| **Activity Variables** |  |  |  |  |  |  |
| Sedentary time (1 hr/day) | 0.493 | 0.299, 0.686 | <0.001 *** | 1.264 | 1.109, 1.441 | <0.001 *** |
| MVPA (10 min/day) | -0.019 | -0.104, 0.065 | 0.651 | 1.006 | 0.951, 1.064 | 0.837 |
| **Covariates** |  |  |  |  |  |  |
| Gender (ref: male) | -0.958 | -1.497, -0.420 | <0.001*** | 0.571 | 0.399, 0.818 | 0.002** |
| Age (yrs) | -0.041 | -0.083, 0.001 | 0.058 | 0.983 | 0.956, 1.010 | 0.212 |
| Education (ref: primary or lower) | -0.015 | -0.539, 0.510 | 0.957 | 0.904 | 0.656, 1.266 | 0.557 |
| Marital status (Married/cohabiting) | -0.221 | -0.745, 0.304 | 0.409 | 0.994 | 0.704, 1.402 | 0.971 |
| Area SES (ref: low neighbourhood SES) | -0.567 | -1.114, -0.024 | 0.041* | 0.861 | 0.630, 1.178 | 0.349 |
| Study site (ref: Hong Kong) | -1.074 | -1.700, -0.448 | <0.001*** | 0.235 | 0.159, 0.346 | <0.001*** |

b = regression coefficient; OR = odds ratio; CIs = confidence intervals; MVPA = moderate-to-vigorous physical activity; SES = socioeconomic status

**Table 3. 2- and 3-way moderating effects on ST and MVPA.**

| | Regression terms | BMI (kg.m$^{-2}$) | | | Being overweight/obese | | |
|---|---|---|---|---|---|---|---|
| | | B | 95% CIs | p-value | OR | 95% CIs | p-value |
| | **2-way interactions** | | | | | | |
| 1 | ST (1 hr/day) * MVPA (10 min/day) | -0.024 | -0.059, 0.012 | 0.189 | 0.996 | 0.974, 1.019 | 0.738 |
| 2 | ST (1 hr/day) * Study site [ref: Hong Kong] | 0.044 | -0.246, 0.333 | 0.768 | 0.991 | 0.816, 1.203 | 0.925 |
| 3 | MVPA (10 min/day) * Study site [ref: Hong Kong] | 0.028 | -0.106, 0.163 | 0.680 | 1.007 | 0.920, 1.102 | 0.880 |
| 4 | ST (1 hr/day) * Food outlet density | -0.0003 | -0.0032, 0.0027 | 0.862 | 0.9987 | 0.9967, 1.0007 | 0.196 |
| 5 | MVPA (10 min/day) *Food outlet density | 0.0007 | -0.0004, -0.0018 | 0.213 | 1.00006 | 0.99926, 1.00087 | 0.878 |
| 6 | ST (1 hr/day) * Gender [ref: male] | 0.093 | -0.201, 0.388 | 0.534 | 1.014 | 0.836, 1.231 | 0.886 |
| 7 | MVPA (10 min/day) * Gender [ref: male] | 0.019 | -0.108, 0.147 | 0.766 | 0.981 | 0.902, 1.066 | 0.651 |
| 8 | ST (1 hr/day) * Neigh. SES [ref: low] | -0.039 | -0.329, 0.251 | 0.792 | 1.095 | 0.904, 1.327 | 0.353 |
| 9 | MVPA (10 min/day) * Neigh. SES [ref: low] | 0.079 | -0.050, 0.208 | 0.229 | 1.034 | 0.951, 1.125 | 0.431 |
| 10 | ST (1 hr/day) * Education [ref: primary or lower] | -0.017 | -0.315, 0.281 | 0.912 | 0.988 | 0.812, 1.203 | 0.906 |
| 11 | MVPA (10 min/day) * Education [ref: primary or lower] | -0.047 | -0.176, 0.082 | 0.474 | 1.002 | 0.922, 1.090 | 0.954 |
| | **3-way interactions** | | | | | | |
| 1 | ST (1 hr/day) * MVPA (10 min/day) * Study site [ref: Hong Kong] | 0.117 | 0.038, 0.196 | 0.004** | 1.077 | 1.015, 1.142 | 0.015* |
| | ST*MVPA interaction in Ghent | 0.044 | -0.017, 0.106 | 0.156 | 1.046 | 0.995, 1.099 | 0.076 |
| | ST @ MVPA mean-1SD | | | | 1.151 | 0.947, 1.399 | 0.156 |
| | ST @ MVPA mean | | | | 1.362 | 1.119, 1.658 | 0.002** |
| | ST @ MVPA mean+1SD | | | | 1.611 | 1.160, 2.238 | 0.004** |
| | ST*MVPA interaction in Hong Kong | -0.073 | -0.123, -0.023 | 0.004** | 0.971 | 0.941, 1.003 | 0.076 |
| | ST @ MVPA mean-1SD | 0.846 | 0.483, 1.210 | <0.001*** | 1.482 | 1.168, 1.880 | 0.001*** |
| | ST @ MVPA mean | 0.573 | 0.310, 0.835 | <0.001*** | 1.329 | 1.119, 1.558 | 0.001*** |
| | ST @ MVPA mean+1SD | 0.299 | 0.026, 0.572 | 0.032* | 1.191 | 0.999, 1.422 | 0.052 |
| 2 | ST (1 hr/day) * Gender [ref: male] * Study site [ref: Hong Kong] | -0.172 | -0.765, 0.420 | 0.567 | 0701 | 0.470, 1.046 | 0.082 |
| 3 | MVPA (10 min/day) * Gender [ref: male] * Study site [ref: Hong Kong] | -0.234 | -0.511, 0.044 | 0.099 | 0.896 | 0.733, 1.096 | 0.285 |
| 4 | ST (1 hr/day) * MVPA (10 min/day) * Gender [ref: male] | -0.016 | -0.089, 0.057 | 0.662 | 1.010 | 0.964, 1.059 | 0.681 |
| 5 | ST (1 hr/day) * Neigh. SES [ref: low] * Study site [ref: Hong Kong] | 0.427 | -0.157, 1.010 | 0.151 | 1.480 | 0.992, 2.208 | 0.055 |
| 6 | MVPA (10 min/day) * Neigh. SES [ref: low] * Study site [ref: Hong Kong] | -0.219 | -0.493, 0.054 | 0.116 | 0.864 | 0.717, 1.041 | 0.125 |
| 7 | ST (1 hr/day) * MVPA (10 min/day) * Neigh. SES [ref: low] | 0.019 | -0.054, 0.093 | 0.604 | 1.022 | 0.974, 1.074 | 0.363 |
| 8 | ST (1 hr/day) * Education [ref: primary or lower] * Study site [ref: Hong Kong] | 0.664 | 0.055, 1.273 | 0.033* | 1.590 | 1.047, 2.414 | 0.029* |
| | ST*Education interaction in Ghent | 0.322 | -0.121, 0.765 | 0.154 | 1.286 | 0.935, 1.769 | 0.121 |
| | ST*Education interaction in Hong Kong | -0.342 | -0.755, -0.060 | 0.100 | 0.809 | 0.610, 1.034 | 0.099 |
| | ST @ Primary or lower education | 0.670 | 0.336, 1.004 | <0.001*** | 1.446 | 1.154, 1.812 | 0.001*** |
| | ST @ Secondary or higher education | 0.328 | 0.025, 0.630 | 0.034* | 1.160 | 0.956, 1.416 | 0.106 |
| 9 | MVPA (10 min/day) * Education [ref: primary or lower] * Study site [ref: Hong Kong] | 0.032 | -0.280, 0.344 | 0.839 | 0.979 | 0.781, 1.228 | 0.856 |
| 10 | ST (1 hr/day) * MVPA (10 min/day) * Education [ref: primary or lower] | -0.064 | -0.138, 0.010 | 0.090 | 0.971 | 0.924, 1.021 | 0.246 |
| 11 | ST (1 hr/day) * Food outlet density (1 km buffer) * Study site [ref: Hong Kong] | -0.0022 | -0.0152, 0.0108 | 0.738 | 0.991 | 0.982, 0.999 | 0.049* |
| | ST*Food outlet density interaction in Ghent | | | | 0.998 | 0.996, 1.000 | 0.082 |
| | ST @ Food outlet density mean-1SD | | | | 1.422 | 1.142, 1.771 | 0.002** |
| | ST @ Food outlet density mean | | | | 1.288 | 1.087, 1.527 | 0.003** |
| | ST @ Food outlet density mean+1SD | | | | 1.167 | 0.970, 1.405 | 0.102 |
| | ST*Food outlet density interaction in Hong Kong | | | | 1.007 | 0.998, 1.113 | 0.111 |
| 12 | MVPA (10 min/day) * Food outlet density (1 km buffer) * Study site [ref: Hong Kong] | 0.0009 | -0.0066, 0.0084 | 0.814 | 1.004 | 0.999, 1.008 | 0.142 |

(*Continued*)

**Table 3.** (Continued)

| | Regression terms | BMI (kg.m$^{-2}$) | | | Being overweight/obese | | |
|---|---|---|---|---|---|---|---|
| | | B | 95% CIs | p-value | OR | 95% CIs | p-value |
| 13 | ST (1 hr/day) * MVPA (10 min/day) * Food outlet density (1 km buffer) | 0.00045 | -0.00033, 0.00123 | 0.257 | 1.00007 | 0.99950, 1.00064 | 0.807 |

BMI = body mass index; *b* = regression coefficient; CIs = confidence intervals; OR = odds ratio; ref = reference category; ST = sedentary time; MVPA = moderate-to-vigorous physical activity; Neigh. SES = neighbourhood socio-economic status

Only significant 3-way interactions (p<0.05) were further investigated by probing 2-way interactions within the third interaction variable, for example, Ghent and Hong Kong for site. Where these 2-way interactions were of at least weak significance (p<0.1), associations were examined to determine any significant differences at different levels of the interaction, e.g., ST associations with MVPA at mean-1SD, mean and mean+1SD.

significantly stronger at lower (i.e., mean -1SD) than higher (i.e., mean +1SD) values of MVPA. A similar effect was observed with respect to being overweight/obese. Hong Kong older adults with low levels of MVPA (e.g., ~18 minutes per day) were 48% more likely to be overweight/obese if they increased their ST by 1 hour a day. In contrast, those engaging in high levels of MVPA (~90 minutes per day) were only 19% more likely to be overweight/obese if they increased their ST by 1 hour per day. In Ghent, an opposite trend was observed, with participants engaging in more MVPA tending to be more susceptible to being overweight/obese as a result of accumulating more ST.

Study site also moderated the interaction between ST and education, whereby the Hong Kong but not the Ghent sample tended to show a two-way ST by education interaction on BMI and being overweight/obese. The positive associations of ST with BMI and being overweight/obese were stronger in Hong Kong older adults with primary or lower education than in those with secondary or higher education.

Finally, the number of food outlets within one kilometre from a resident's home tended to moderate the association between ST and being overweight/obese in Ghent, but not in Hong Kong. Associations of ST with being overweight/obese in Ghent were weaker in participants with a higher food outlet density in their neighbourhood than in those with a lower food outlet density.

## Discussion

This observational study, conducted in two culturally and geographically very diverse regions, aimed to examine the main and interaction effects of objectively-assessed ST and MVPA with weight outcomes in older adults. Furthermore, potential moderating effects of gender, educational level, food outlet density and neighbourhood-level SES were studied. Finally, the generalizability of all associations and interactions across study sites was examined. This study revealed some interesting and surprising findings, which are summarized and discussed below.

ST was linearly and positively related to BMI and the odds of being overweight or obese in the overall sample, while MVPA was not. The overall-sample analyses including the two-way interaction between MVPA and ST showed no interactions between these behaviours on weight outcomes. Furthermore, none of the other two-way interactions were significant, meaning that the included socio-demographic characteristics and objective neighbourhood characteristics did not moderate the associations of MVPA, ST with the weight outcomes. So, in the overall sample, the non-significant association of MVPA as well as the linear positive association of ST with BMI and being overweight or obese, were generalisable across genders, educational levels, neighbourhood-level SES and across different densities of food outlets in

the neighbourhood. However, an interesting and remarkable finding was that three site-specific findings were identified, confirming the complexity of the relation between PA, ST and weight outcomes.

The absence of a main effect of MVPA on weight outcomes in this sample of Hong Kong and Ghent older adults is surprising. Many other cross-sectional and longitudinal studies in older adults showed that MVPA is associated with a lower risk of overweight and obesity [15,49,50]. No definite explanation for our finding can be given, but it is important to note that the participants in both study sites were quite active. A large proportion of the older adults, both in Hong Kong (90.0%) and in Ghent (68.9%) reached the health guideline of 150 minutes of MVPA per week. This limited variability in MVPA might have contributed to not finding an association with the weight outcomes. Furthermore, other accelerometer cut-points were used to define MVPA in previous studies [15,50], possibly leading to different associations.

The robust association found between ST and the weight outcomes, independent of the levels of PA, is promising. When confirmed in future studies with a stronger study design, this may imply that a focus on limiting ST as a preventive strategy against overweight and obesity could be effective in older adults, especially in those who are physically incapable of being highly active. Naturally, in light of the co-dependency of PA, ST and sleep within a 24-hour cycle, it is necessary to determine by which behaviour ST ideally should be replaced. Recent evidence from studies examining the theoretical [51] or real-life [50] effect of behavioural substitution in older adults, shows that reallocation of ST to different intensities of PA can have a positive effect on obesity markers. However, current evidence is strongest for reallocation to MVPA. Dumuid and colleagues [51] found a theoretical decrease of -0.7 BMI points if 15 minutes of ST were replaced by 15 minutes of MVPA, and concluded that reallocation of ST to light-intensity PA had no impact on older adults' BMI. In contrast, a seven-year longitudinal study in older women concluded that an increase in light-intensity PA or MVPA at the expense of ST was related to a lower BMI and percentage body fat [50]. However, also in that study, the largest change in BMI and percentage body fat was found when sedentary behaviour was longitudinally replaced by MVPA. More longitudinal studies in larger and more diverse groups of older adults are needed before concrete recommendations can be given regarding replacing ST by light- or moderate-intensity PA to prevent overweight or obesity.

Interestingly, although main effects of ST on the weight outcomes were identified in the total sample, site-specific moderating effects of educational level and food outlet density were revealed. In Hong Kong, the association between ST and the weight outcomes was stronger in lower-educated than in higher-educated participants, while no moderating effect of educational level was found in Ghent. In general, lower-educated individuals often have a less healthy lifestyle than their higher-educated counterparts [52,53]. Thus, it is possible that higher-educated Hong Kong older adults might have compensated for the negative effects of ST by following a healthier diet. In fact, educational attainment was one of the two strongest predictors of dietary quality in Chinese older adults [54]. Lack of nutritional knowledge, lack of cooking skills and apathy toward nutrition messages have been identified as potential reasons for the unhealthy diet among educationally and economically disadvantaged people [55]. The fact that educational attainment was not a significant moderator of the ST-weight outcomes associations in Ghent may be due to the much higher level of education among participants from Ghent (~75% with above primary education) than from Hong Kong (~47% with above primary education). In Ghent, a positive association between ST and being overweight/obese was found in participants with a low or mean food outlet density in their neighbourhood and not in those with a high food outlet density. This seems to be logical, as limited access to healthy food options has been associated with less healthy dietary intake, which is in turn

associated with higher odds of being overweight or obese [27,28]. The fact that no such moderating effect was found in Hong Kong might be due to Hong Kong residents living in neighbourhoods with few food outlets being able to easily access via public transport destination-rich neighbourhoods with many food options [56].

A site-specific interaction effect of PA and ST on BMI was identified. In Hong Kong, the positive association between ST and BMI was stronger in older adults with lower levels of MVPA than in those with higher levels of MVPA. In contrast, in Ghent, the positive association between ST and the odds of being overweight or obese was unexpectedly stronger in older adults with higher instead of lower levels of MVPA. This suggests that only in Hong Kong older adults, higher levels of MVPA seem to be protective against the positive association between ST and BMI. In Ghent, no such potentially protective effect of MVPA was identified, even a reverse ('unprotective') association was found. Overall, all these moderating effects confirm the complexity of some of the examined associations. It can be cautiously suggested that, although an overall association between ST and weight outcomes was found, strategies to limit ST might have stronger effects in specific subgroups located in specific regions (e.g. lower-educated adults in Hong Kong, those living in neighbourhoods with low food outlet density in Ghent). Consequently, strategies to prevent overweight and obesity through a focus on ST might not be equally effective across regions. Furthermore, the results suggest that unassessed country-specific variables (e.g. dietary patterns, perceptions of the built environment) may play a key confounding role. Future international studies combining data of multiple geographically and culturally-diverse countries are strongly encouraged to further disentangle these complex associations.

Finally, potential curvilinearity of the relationships of PA and ST with weight outcomes was examined. A previous multi-country study in adults found a curvilinear association of MVPA with BMI and the odds of being overweight or obese, where a linear negative relation levelled off at higher levels of MVPA [24]. No such associations were found here. The relation between ST and weight status was consistently linear, implying that higher levels of ST are consistently related to higher levels of BMI and higher odds of overweight and obesity. If confirmed in future studies also taking into account potential curvilinearity in their analyses, this may help to shape recommendations regarding limiting ST in older adults in order to prevent overweight or obesity.

A first strength of this study was the objective assessment of PA and ST. Most previous studies only included self-reported measures of ST, often limited to TV viewing [15]. Second, weight and height were also objectively assessed in both samples. It is known that individuals (mainly those with overweight or obesity) tend to underreport their weight when self-assessment is used [57]. Third, comparable study protocols were used in two culturally and geographically diverse countries. This study also has some weaknesses. First, this was an observational, cross-sectional study, precluding any conclusions regarding causality of the results. Second, mainly active older adults participated in the study. Since study participation was voluntarily, it is possible that mainly motivated and active older adults consented to participate, especially in Ghent, which had a lower response rate than Hong Kong. This may have influenced the findings and limits the generalisability of the current results. Third, although it is known that dietary intake is an important precursor of overweight and obesity, dietary intake was not assessed in this study. Two proxy measures (neighbourhood-level SES and food outlet density) that have been associated with (un)healthy food intake were included as moderators, but the study would have been stronger if dietary intake had been assessed directly. Finally, although accelerometers measure PA and ST objectively, only the behaviour in one specific week was assessed. This specific week might not be representative of a habitual week

(e.g. due to bad weather or physical complaints during that week), and this might have biased the results.

## Conclusion

In conclusion, this study showed that ST was linearly and positively related to BMI and the odds of being overweight/obese in Hong Kong and Belgian older adults, while MVPA was not. This finding was generalizable across men and women, high- and low-educated individuals, those living in high- and low-SES neighbourhoods and with high and low access to healthy food options. If confirmed in future longitudinal studies, it can be carefully suggested that a focus on limiting ST as a preventive strategy against overweight and obesity might be effective in older adults, especially in those physically incapable of being very active. However, some country-specific effects were found, confirming the cultural dependency and complexity of the associations between PA, ST and weight outcomes. Future longitudinal international studies including older adults from multiple geographically and culturally-diverse regions assessing PA, ST, weight outcomes and dietary intake should be encouraged to further disentangle the complex associations. Such studies are needed to refine the recommendations regarding ST and PA in older adults in light of preventing overweight and obesity.

## Supporting information

**S1 Data.**
(SAV)

## Author Contributions

**Conceptualization:** Delfien Van Dyck, Anthony Barnett, Jelle Van Cauwenberg, Casper J. P. Zhang, Cindy H. P. Sit, Ester Cerin.

**Data curation:** Ester Cerin.

**Formal analysis:** Anthony Barnett, Ester Cerin.

**Funding acquisition:** Delfien Van Dyck, Jelle Van Cauwenberg, Cindy H. P. Sit, Ester Cerin.

**Investigation:** Delfien Van Dyck, Casper J. P. Zhang.

**Methodology:** Delfien Van Dyck, Jelle Van Cauwenberg, Ester Cerin.

**Project administration:** Delfien Van Dyck, Jelle Van Cauwenberg.

**Supervision:** Delfien Van Dyck, Jelle Van Cauwenberg, Ester Cerin.

**Writing – original draft:** Delfien Van Dyck, Anthony Barnett, Ester Cerin.

**Writing – review & editing:** Delfien Van Dyck, Anthony Barnett, Jelle Van Cauwenberg, Casper J. P. Zhang, Cindy H. P. Sit, Ester Cerin.

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
