## [Decision Letter · Decision Letter 0]

12 May 2020

PONE-D-20-03534

Main and interacting effects of physical activity and sedentary time on older adults’ BMI: the moderating roles of socio-demographic and environmental attributes

PLOS ONE

Dear Dr. Van Dyck,

Thank you for submitting your manuscript to PLOS ONE. After careful consideration, we feel that it has merit but does not fully meet PLOS ONE’s publication criteria as it currently stands. Therefore, we invite you to submit a revised version of the manuscript that addresses the points raised during the review process.

We would appreciate receiving your revised manuscript by Jun 26 2020 11:59PM. To enhance the reproducibility of your results, we recommend that if applicable you deposit your laboratory protocols in protocols.io, where a protocol can be assigned its own identifier (DOI) such that it can be cited independently in the future. For instructions see: http://journals.plos.org/plosone/s/submission-guidelines#loc-laboratory-protocols

We look forward to receiving your revised manuscript.

Kind regards,

Martin Senechal, PhD

Academic Editor

PLOS ONE

Reviewers' comments:

Reviewer's Responses to Questions

**Comments to the Author**

1. Is the manuscript technically sound, and do the data support the conclusions?

Reviewer #1: Yes

Reviewer #2: Yes

2. Has the statistical analysis been performed appropriately and rigorously? 

Reviewer #1: I Don't Know

Reviewer #2: Yes

3. Have the authors made all data underlying the findings in their manuscript fully available?

Reviewer #1: No

Reviewer #2: Yes

4. Is the manuscript presented in an intelligible fashion and written in standard English?

Reviewer #1: Yes

Reviewer #2: Yes

5. Review Comments to the Author

Reviewer #1: Review of the manuscript

Main and interacting effects of physical activity and sedentary time on older adults’ BMI: the moderating roles of socio-demographic and environmental attributes.

General comments:

The manuscript is interesting! The authors have taken already sampled data and made new use of the data to evaluate the relationship between MVPA and sedentary activity.

Methods and material:

Good methods, well described, and useable, I only have a few comments and questions.

The authors are using Actigraphs, and describe the cut points for MVPA and sedentary activity, they have defined non-wear time as 90 minutes? Is there a reference for this, it seems a bit high, personally I would have chosen 60 minutes, but it could be because older adults are less agile might be very still over longer time span than younger people, therefore it might be OKAY, but I would like the authors to use 60 minutes instead of 90 minutes to see if it changes the accelerometer wear time and thereby the results of their analyzes, they might have to exclude some participants as they do not get time enough to be used in the analyzes.

Data analyzes are performed using a method that is not used often, but seems to be good for the objective. But when the authors use a rarely used method, I would like to explain why they are using GAMM, from how I understand the literature GAMM are usually used for longitudinal data as GAMM are especially good for estimating trend see e.g. [1]. So please explain why you chose to use GAMM, and what the advantage is compared to e.g. GEE or mixed linear models.

Minor problem: line 235, you cite Wood, but without a reference number.

Results:

It seems that the authors are using a kind of substitution model, but their analyzes are difficult to follow. They state that with lower amount of MVPA then an increase in sedentary activity had a higher likelihood of generating increased BMI or obesity. But as amount of Sedentary activity is highly associated with MVPA (negatively) and including both in the same analysis could be problematic. The participants “only” have 100% time or 24 hour a day, now if they increase ST then they have to reduce either MVPA or low activity, thus when the authors make an assumption of increase of both MVPA and ST activity, it can only come from low activity, now how likely is to increase both ST and MVPA? To my knowledge this has been addressed by Heidemann et al [2] see https://www.ncbi.nlm.nih.gov/pubmed/23452342, the approach will enable a graphical illustration that will it easier to understand the authors point.

Discussion:

Short and to the point, good.

I miss a mentioning or discussion of one important weakness of accelerometry, we measure for a given time, and have the assumption that the activity we measure is the habitual activity! This might not be so, we know that weather and seasons have a large impact on physical activity, so the authors need to address this weakness, al though I do agree that it is a strength they have been using accelerometers rather than questionnaires.

Conclusion of review: Interesting cross sectional study that raises new questions and hypotheses, with a few additions and changes it should be ready for publication.

1. Shadish WR, Zuur AF, Sullivan KJ: Using generalized additive (mixed) models to analyze single case designs. J Sch Psychol 2014, 52(2):149-178.

2. Heidemann M, Molgaard C, Husby S, Schou AJ, Klakk H, Moller NC, Holst R, Wedderkopp N: The intensity of physical activity influences bone mineral accrual in childhood: the childhood health, activity and motor performance school (the CHAMPS) study, Denmark. BMC pediatrics 2013, 13:32.

Reviewer #2: In the current manuscript, Dyck et al assessed the main and interacting effects of PA and ST with BMI and overweight/obese in Hong Kong and Ghent older adults. They found that ST was linearly and positively related to weight outcomes in Hong Kong and Belgian older adults, while MVPA was not. Although this study have important public health implications, but there are some limitations to the study. The manuscript could be strengthened by addressing the following issues and then reconsidering the potential for publication:

1. The author found the positive association between ST and BMI was significantly stronger at lower than higher values of MVPA in Hong Kong older adults but an opposite trend in Belgian population. Table 1 should compares baseline characteristics of the study participants from Hong Kong and those from Ghent using relevant statistical methods. Knowing differences between these groups helps the reader understand the results of this article.

2. What is the explanation for why the study participants in Hong Kong were mostly women?

3. The author adopted a targeted obesity standard for the Hong Kong population, is there a targeted obesity standard for the Belgian population?

4. A large proportion of the older adults, both in Hong Kong (90.0%) and in Ghent (68.9%) reached the health guideline of 150 minutes of MVPA per week. The author could divide these participants into moderate physical activity group and heavy physical activity group to further explore the association of ST and PA with weight outcomes.

5. Please rephrase with a focus on the public health implications of your findings in terms of its practical utility taking current practices and clinical feasibility into account in Conclusions.

6. PLOS authors have the option to publish the peer review history of their article (what does this mean?). If published, this will include your full peer review and any attached files.

Reviewer #1: Yes: Niels Wedderkopp

Reviewer #2: No

---

## [Author Response · Author response to Decision Letter 0]

11 Jun 2020

Comments of reviewer #1

Main and interacting effects of physical activity and sedentary time on older adults’ BMI: the moderating roles of socio-demographic and environmental attributes.

General comments:

The manuscript is interesting! The authors have taken already sampled data and made new use of the data to evaluate the relationship between MVPA and sedentary activity.

Thank you for your positive evaluation of the manuscript. 

Methods and material:

Good methods, well described, and useable, I only have a few comments and questions.

1. The authors are using Actigraphs, and describe the cut points for MVPA and sedentary activity, they have defined non-wear time as 90 minutes? Is there a reference for this, it seems a bit high, personally I would have chosen 60 minutes, but it could be because older adults are less agile might be very still over longer time span than younger people, therefore it might be OKAY, but I would like the authors to use 60 minutes instead of 90 minutes to see if it changes the accelerometer wear time and thereby the results of their analyses, they might have to exclude some participants as they do not get time enough to be used in the analyses.

In the manuscript, we refer to the paper of Choi and colleagues (reference 38) to justify our decision to define accelerometer non-wear time as >90 minutes of consecutive zeros. In that paper, the accuracy of accelerometer wear/non-wear classification in older adults was examined by using two time-windows (60 minutes and 90 minutes). Both wrist-worn and waist-worn Actigraph accelerometers were used in that study. Results showed that the 90-minute time window was more accurate than the 60-minute time window to classify wear/non-wear time in free-living older adults. Based on findings from validation studies in older adults, other researchers have used even higher cut-off values to determine non-wear time in older adults, such as >100 minutes of consecutive zeros (Cerin et al., 2016; Davis et al., 2011; Davis and Fox, 2007). 

We agree that a 60-minute window is more optimal for use in a general adult population, but because of the fact that older adults indeed may be sedentary (but still wearing the accelerometer) for longer periods, using 60 minutes as a cut-off value does not seem to be optimal in this age group. Consequently, based on the results of the study of Choi et al, we decided to keep this cut-off value. 

Cerin E, Zhang CJ, Barnett A, et al. Associations of objectively-assessed neighborhood characteristics with older adults' total physical activity and sedentary time in an ultra-dense urban environment: Findings from the ALECS study. Health Place. 2016;42:1‐10. 

Choi L, Ward SC, Schnelle JF, Buchowski MS. Assessment of wear/nonwear time classification algorithms for triaxial accelerometer. Med Sci Sports Exerc. 2012; 44: 2009-2016. 

Davis MG, Fox KR, Hillsdon M, Sharp DJ, Coulson JC, Thompson JL. Objectively measured physical activity in a diverse sample of older urban UK adults. Med Sci Sports Exerc. 2011;43(4):647‐654.

Davis MG, Fox KR. Physical activity patterns assessed by accelerometry in older people. Eur J Appl Physiol. 2007;100(5):581‐589. 

2. Data analyses are performed using a method that is not used often, but seems to be good for the objective. But when the authors use a rarely used method, I would like to explain why they are using GAMM, from how I understand the literature GAMM are usually used for longitudinal data as GAMM are especially good for estimating trend see e.g. [1]. So please explain why you chose to use GAMM, and what the advantage is compared to e.g. GEE or mixed linear models.

Generalized Additive Mixed Regression Models (GAMMs) were chosen because of the following reasons:

- mixed models can account for multiple levels of clustering (e.g. individuals in neighbourhoods in countries)

- generalized models can model non-normally distributed outcomes like physical activity

- additive models can explore the shape (e.g. curvilinearity) of the associations between explanatory variables and outcomes. 

So, the added value of using GAMMs instead of linear mixed models is mainly the fact that potential non-linearity of associations can be examined, and that non-normally distributed outcomes can be included. Also compared to GEE models, GAMMs have the added value that they can identify non-linear associations. 

In the Data Analyses section we explain the added value of using GAMMs (page 10). 

“To account for clustering at the administrative unit level and the possibility of curvilinear associations, generalised additive mixed models (GAMMs) [44] were used…”

3. Minor problem: line 235, you cite Wood, but without a reference number.

Thank you for noticing this, the reference number for Wood, 2006 is [44]. We changed this in the manuscript. 

4. Results:

It seems that the authors are using a kind of substitution model, but their analyses are difficult to follow. They state that with lower amount of MVPA then an increase in sedentary activity had a higher likelihood of generating increased BMI or obesity. But as amount of Sedentary activity is highly associated with MVPA (negatively) and including both in the same analysis could be problematic. The participants “only” have 100% time or 24 hour a day, now if they increase ST then they have to reduce either MVPA or low activity, thus when the authors make an assumption of increase of both MVPA and ST activity, it can only come from low activity, now how likely is to increase both ST and MVPA? To my knowledge this has been addressed by Heidemann et al [2] see https://www.ncbi.nlm.nih.gov/pubmed/23452342, the approach will enable a graphical illustration that will it easier to understand the authors point.

To explain our analyses, we would like to refer to our answer to comment 2 of this reviewer. We did not use a substitution model, but regression models (GAMMs) to examine the associations of MVPA and ST with BMI and the odds of being overweight/obese. Furthermore, because the data were not longitudinal, nor did we use a substitution model, we cannot (and did not) make statements about ‘increases’ or ‘decreases’ in ST or PA, but only about ‘higher’ and ‘lower’ levels of these behaviours. So, we did not state that an ‘increase’ in ST leads to a higher likelihood of overweight/obesity. What our findings do show, is that in Hong Kong older adults, there was an interaction between ST and MVPA to explain BMI and being overweight/obese. So, higher levels of ST were associated with a higher BMI, and the association was strongest in individuals with lower values of MVPA. In Ghent, an opposite trend was revealed, with stronger associations between ST and BMI in those older adults with higher levels of MVPA. Based on these results, we do not assume an increase of both MVPA and ST, and nowhere in the Discussion section did we suggest this. We do mention that both MVPA and ST can be high in individuals. This has been shown in previous research as well, and is called the ‘active couch potato’ phenomenon (Owen et al, 2010). 

Nonetheless, we do agree that individuals only have 24 hours in a day, and that PA, ST and sleep are co-dependent. Consequently, we based our suggestions in the Discussion on these assumptions. On page 20, we suggest that limiting ST as a preventive strategy against overweight and obesity could be effective in older adults. Furthermore, we suggest that reallocation of ST to PA of different intensities (light-intensity and MVPA) can have a positive effect on obesity markers, with evidence being strongest for MVPA. This suggestion is based on existing literature, mentioned on page 20 of the manuscript. 

We hope that this explanation clarifies the Results for reviewer #1. 

Owen N, Healy GN, Matthews CE, Dunstan DW. Too much sitting: the population health science of sedentary behavior. Exerc Sport Sci Rev 2010; 38: 105-113. 

5. Discussion:

Short and to the point, good. I miss a mentioning or discussion of one important weakness of accelerometry, we measure for a given time, and have the assumption that the activity we measure is the habitual activity! This might not be so, we know that weather and seasons have a large impact on physical activity, so the authors need to address this weakness, although I do agree that it is a strength they have been using accelerometers rather than questionnaires.

We added this to the Limitations on page 24. 

“Finally, although accelerometers measure PA and ST objectively, only the behaviour in one specific week was assessed. This specific week might not be representative of a habitual week (e.g. due to bad weather or physical complaints during that week), and this might have biased the results.”

Conclusion of review: Interesting cross sectional study that raises new questions and hypotheses, with a few additions and changes it should be ready for publication.

Thank you once again for your positive and helpful feedback. 

1. Shadish WR, Zuur AF, Sullivan KJ: Using generalized additive (mixed) models to analyze single case designs. J Sch Psychol 2014, 52(2):149-178.

2. Heidemann M, Molgaard C, Husby S, Schou AJ, Klakk H, Moller NC, Holst R, Wedderkopp N: The intensity of physical activity influences bone mineral accrual in childhood: the childhood health, activity and motor performance school (the CHAMPS) study, Denmark. BMC pediatrics 2013, 13:32.

Comments of reviewer #2

In the current manuscript, Dyck et al assessed the main and interacting effects of PA and ST with BMI and overweight/obese in Hong Kong and Ghent older adults. They found that ST was linearly and positively related to weight outcomes in Hong Kong and Belgian older adults, while MVPA was not. Although this study have important public health implications, but there are some limitations to the study. The manuscript could be strengthened by addressing the following issues and then reconsidering the potential for publication.

We would like to thank reviewer 2 for his/her positive evaluation of the manuscript. 

1. The author found the positive association between ST and BMI was significantly stronger at lower than higher values of MVPA in Hong Kong older adults but an opposite trend in Belgian population. Table 1 should compares baseline characteristics of the study participants from Hong Kong and those from Ghent using relevant statistical methods. Knowing differences between these groups helps the reader understand the results of this article.

We have now reported the results of a formal comparison of baseline characteristics between Hong Kong and Ghent participants in Table 1. However, we wish to note that these analyses cannot explain the observed between-sample differences in moderating effects of MVPA on the association between ST and BMI. In fact, the estimates of the moderating effect of MVPA were adjusted for all baseline characteristics. On page 10, we have now stated:

“Descriptive statistics of sample characteristics (participant demographics, objective neighbourhood attributes, activity levels, and BMI) were computed. We also tested the significance (p<0.05) of the differences in these characteristics between the two samples using, as appropriate, generalised linear models and multinomial logistic regression with robust standard errors accounting for clustering at the administrative unit level.”

Furthermore, we added the baseline comparisons (p-values) in the Results (page 11). 

 “Participant characteristics are presented in Table 1. Mean age was 75 years and slightly higher in the Hong Kong sample (p=0.025). The total sample (n=829) consisted of 61% female participants, 69% and 54% in Hong Kong and Ghent, respectively (p<0.001). The sample from Ghent had a higher level of education than the Hong Kong sample (p<0.001), with 75% versus 47% having an educational level higher than primary school. Having a car in the household was much more prevalent in Ghent (81%) than in Hong Kong (29%) (p<0.001). The mean ST per day was 414 minutes, with the Ghent cohort accumulating an average of 16 more minutes per day than the Hong Kong cohort (p=0.795). For the total cohort, the average MVPA was 56 minutes per day, with Hong Kong participants engaging in approximately 26 minutes more per day than their Ghent counterparts (p<0.001). Mean BMI was 23 kg/m2 and, on average, ~ 1kg/m2 lower in Ghent (p<0.001). Because of the different cut-off values for categorising overweight/obesity in Asian and European older adults [33], the distribution of BMI categories was very different across sites with 57% of those in Hong Kong being overweight/obese as compared to 24% in Ghent (p<0.001).”

2. What is the explanation for why the study participants in Hong Kong were mostly women?

The Hong Kong sample consisted of substantially more women than men for two reasons. First, there are more women than men among older Hong Kong Chinese 65 years and over (54%; Census and Statistics Department, Government of HK SAR, 2020). Second, men were less likely to consent participating in the study. Of the 29% potential participants who refused to join the study, approximately 70% were men. This is not unusual as many studies have found that women are more likely to participate in studies than men (e.g., Moore and Tarnai, 2002; Singer et al., 2000) 

Census and Statistics Department, Hong Kong SAR. Populations estimates: Table 002 – Population by age group and sex. Available at: https://www.censtatd.gov.hk/hkstat/sub/sp150.jsp?tableID=002&ID=0&productType=8 (retrieved: 30 May 2020).

Moore DL, Tarnai J. Evaluating nonresponse error in mail surveys. In: Groves RM, Dillman DA, Eltinge JL, Little RJA (eds.), Survey Nonresponse. New York: John Wiley & Sons, 2002. (pp. 197–211).

Singer E, van Hoewyk J, Maher MP. Experiments with incentives in telephone surveys. Public Opinion Quarterly. 2000;64:171–188. 

3. The author adopted a targeted obesity standard for the Hong Kong population, is there a targeted obesity standard for the Belgian population?

No, for the Belgian population, the WHO standards to define overweight and obesity were used (https://gateway.euro.who.int/en/indicators/mn_survey_19-cut-off-for-bmi-according-to-who-standards/). According to these cut offs, a BMI score of ≥25 kg/m² is defined as overweight and a BMI score of ≥30 kg/m² is defined as obesity. These WHO standards are used globally, so there was no need to adapt these for Belgian older adults. 

However, as noted in the paper (WHO Expert Consultation, 2004), appropriate BMI cut offs for Asian populations have been developed and the WHO recommends to use these cut offs in research with Asian populations. Therefore, we used these adjusted cut offs to define overweight in the Hong Kong sample. 

WHO Expert Consultation. Appropriate body-mass index for Asian populations and its implications for policy and intervention strategies. Lancet. 2004; 363: 157-63. Erratum in: Lancet. 2004; 363: 902.

4. A large proportion of the older adults, both in Hong Kong (90.0%) and in Ghent (68.9%) reached the health guideline of 150 minutes of MVPA per week. The author could divide these participants into moderate physical activity group and heavy physical activity group to further explore the association of ST and PA with weight outcomes.

This is an interesting suggestion. However, we used the cut points of Barnett et al (2016) to define MVPA in our sample of older adults. Barnett and colleagues developed accelerometer-based MVPA cut points for older adults. They did not develop separate cut points for moderate and for vigorous PA. Consequently, we cannot divide the current sample into a moderate and a high-intensity PA group, to further explain the associations of ST and PA with weight outcomes. 

We chose the cut points of Barnett and colleagues for several reasons. 

1) They were developed specifically for older adults, while most other cut points were developed for adults. The resting metabolic rate of older adults is lower than of adults, so it is important to take this into account in calibration studies. 

2) Other studies determining Actigraph cut points in older adults included other activities like household chores or light intensity exercises that typically occur infrequently in real life. Walking is a preferred activity for older adults to increase their levels of PA and is therefore the most suited activity to determine accelerometer-based cut points. 

3) Overground walking was used instead of treadmill walking. When treadmill walking is used to determine cut points, ActiGraph counts are lower and energy expenditure is higher for a given walking speed compared with overground walking. Consequently, energy expenditure walking speed predicted by the accelerometer are overestimated in ream-life conditions. 

Furthermore, according to studies that used ActiGraph accelerometers to measure PA, older adults tend to accumulate very minimal amounts of vigorous-intensity PA. For example, in a study conducted in Hong Kong, older adults accumulated, on average, only 1 minute (SD = 2.0) of vigorous-intensity PA in a week (Cerin et al., 2012). Similarly, a study conducted in Sweden found that the median daily minutes of accelerometry-assessed vigorous-intensity PA in older adults was 0 (Hurtig-Wennlöf et al., 2010). 

For these reasons, we decided to use the cut points of Barnett and colleagues, and a distinction between moderate and vigorous PA cannot be made. 

Barnett A, Van den Hoek D, Barnett D, Cerin E. Measuring moderate-intensity walking in older adults using the Actigraph accelerometer. BMC Geriatrics. 2016; 16: 211. 

Cerin E, Barnett A, Cheung MC, Sit C HP, Macfarlane DJ, Chan WM. Reliability and validity of the IPAQ-L in a sample of Hong Kong urban older adults: does neighborhood of residence matter?. J Aging Phys Act. 2012;20(4):402‐420. 

Hurtig-Wennlöf A, Hagströmer M, Olsson LA. The International Physical Activity Questionnaire modified for the elderly: aspects of validity and feasibility. Public Health Nutr. 2010;13(11):1847‐1854.

5. Please rephrase with a focus on the public health implications of your findings in terms of its practical utility taking current practices and clinical feasibility into account in Conclusions.

In the previous version of the manuscript, we indeed formulated a rather careful conclusion because this study is an observational study, so no assumptions regarding causality of the associations can be made. Furthermore, to our knowledge, this was one of the first studies examining the interaction effects between ST and MVPA on weight outcomes in older adults, so future studies should confirm our findings before firm recommendations can be made.

However, based on our findings (and like we did in the Discussion) we can carefully suggest that a focus on limiting ST as a preventive strategy against overweight and obesity may be effective in older adults, especially in those who are physically incapable of being highly active. Ideally, the ST should then be replaced by MVPA, but replacing it by light-intensity PA might already be beneficial. We added this to the Conclusion of the paper, but we did not add the suggestion regarding replacement by MVPA or light-intensity PA, because we did not examine this ourselves.

“In conclusion, this study showed that ST was linearly and positively related to BMI and the odds of being overweight/obese in Hong Kong and Belgian older adults, while MVPA was not. This finding was generalizable across men and women, high- and low-educated individuals, those living in high- and low-SES neighbourhoods and with high and low access to healthy food options. If confirmed in future longitudinal studies, it can be carefully suggested that a focus on limiting ST as a preventive strategy against overweight and obesity might be effective in older adults, especially in those physically incapable of being very active.”

---

## [Decision Letter · Decision Letter 1]

24 Jun 2020

Main and interacting effects of physical activity and sedentary time on older adults’ BMI: the moderating roles of socio-demographic and environmental attributes

PONE-D-20-03534R1

Dear Dr. Van Dyck,

We’re pleased to inform you that your manuscript has been judged scientifically suitable for publication and will be formally accepted for publication once it meets all outstanding technical requirements.

Kind regards,

Martin Senechal, PhD

Academic Editor

PLOS ONE

Additional Editor Comments (optional):

Reviewers' comments:

Reviewer's Responses to Questions

**Comments to the Author**

1. If the authors have adequately addressed your comments raised in a previous round of review and you feel that this manuscript is now acceptable for publication, you may indicate that here to bypass the “Comments to the Author” section, enter your conflict of interest statement in the “Confidential to Editor” section, and submit your "Accept" recommendation.

Reviewer #1: (No Response)

Reviewer #2: All comments have been addressed

2. Is the manuscript technically sound, and do the data support the conclusions?

Reviewer #1: Yes

Reviewer #2: Yes

3. Has the statistical analysis been performed appropriately and rigorously? 

Reviewer #1: Yes

Reviewer #2: Yes

4. Have the authors made all data underlying the findings in their manuscript fully available?

Reviewer #1: No

Reviewer #2: Yes

5. Is the manuscript presented in an intelligible fashion and written in standard English?

Reviewer #1: Yes

Reviewer #2: Yes

6. Review Comments to the Author

Reviewer #1: The Authors have generally followed the advice form the reviewers, and if not have a good and valid explanation for why not. In my opinion the manuscript should be ready for publication.

Reviewer #2: (No Response)

7. PLOS authors have the option to publish the peer review history of their article (what does this mean?). If published, this will include your full peer review and any attached files.

Reviewer #1: **Yes: **Professor Niels Wedderkopp

Reviewer #2: No

---

## [Editor Report · Acceptance letter]

30 Jun 2020

PONE-D-20-03534R1 

Main and interacting effects of physical activity and sedentary time on older adults’ BMI: the moderating roles of socio-demographic and environmental attributes 

Dear Dr. Van Dyck:

I'm pleased to inform you that your manuscript has been deemed suitable for publication in PLOS ONE. Congratulations! Your manuscript is now with our production department. 

Kind regards, 

on behalf of

Dr. Martin Senechal 

Academic Editor

PLOS ONE